# The portrayal of panic-buying and stockpiling in English newspapers during Covid, a mixed-method content analysis

Dayna Brackley ⓘ *, Rebecca Wells ⓘ

Centre for Food Policy, City St George's, University of London, London, United Kingdom

* dayna@bremnerco.com

## Abstract

Panic-buying and stockpiling during Covid disrupted the supply chain, causing food shortages and impacting the vulnerable. The government faced criticism for its lack of food system resilience, poor communications planning, and reliance on retailers. The British media frequently reported on panic-buying during early lockdown stages in 2020 and throughout Covid. The media play an important role in communicating information to the British public during times of crisis and influence public opinion. This mixed-method study examined English media portrayal of panic-buying, analysing text and visual data from six of the highest-circulating newspapers from March to July 2020. It reviewed reporting trends, use of imagery, themes, and prominent stakeholder voices. Content analysis of 209 articles showed that coverage was dominated by popular and left-wing press, with 89% of articles using sensationalised language and 68% coded as negative. In a subset of 125 articles, visual imagery showed empty shelves in 64% of analysed images, reinforcing the impression of food shortages. Supermarkets were the most quoted stakeholders, appearing in 62% of articles. Contradictions included reports of no food shortages alongside images of empty shelves and early newspaper advice encouraging stockpiling. Reporting peaked between March 16–22, 2020. Six key themes were identified: supermarket prominence, food supply/access, food policy, individual behaviour, socio-economic impacts, and panic-buying drivers—all themes had relevance to food system resilience. Future civil unrest linked to food-system challenges, potentially driven by climate change, conflict, or political instability, could see panic-buying play a significant role. Research on media portrayals of panic-buying can help policymakers enhance communication strategies and identify critical issues during crises. The Covid pandemic revealed crucial lessons about the media's potential role in shaping public behaviour, highlighting the need for stronger government communication and collaboration with both the media and retailers to ensure consistent messaging, particularly to protect vulnerable groups.

**Data Availability Statement:** All relevant data are within the manuscript and its Supporting information files.

**Funding:** The author(s) received no specific funding for this work.

**Competing interests:** The authors have declared
that no competing interests exist.

## Introduction

This paper explores the media's role in shaping public perceptions of panic-buying during the
Covid pandemic in England. By analysing content from prominent newspapers, this study
explores how media coverage may have influenced public behaviours, exacerbated panic-buy-
ing, and highlighted weaknesses in food system resilience. This paper also examines the impli-
cations of these media narratives for public policy and crisis management in future
disruptions.

### Background and context

The World Health Organization declared Covid a pandemic on March 11, 2020, and the
United Kingdom (UK) government imposed a national lockdown on March 23, 2020 [1, 2].
People were asked to stay home, with only essential outings allowed. All restaurants, pubs and
bars closed, except for take-away outlets [3, 4]. These measures significantly impacted the food
system, resulting in widespread panic-buying and stockpiling [5]. In the week before lock-
down, supermarkets saw record sales [3, 6]. Panic-buying, the hoarding of household goods
due to scarcity or fear of shortages [7], was widespread during the pandemic, exacerbating
food supply chain issues.

### Government and retail response to panic-buying

Panic-buying was identified as a risk during no-deal Brexit preparations [8, 9]. During Covid,
the government was criticised for relying on supermarkets, poor communication, and inade-
quate recognition of the food supply chain [10–15]. The UK's just-in-time supply chain strug-
gled with fluctuating demand [10, 16] and consumers focused on long shelf-life products amid
shortages [17–20]. 83% of consumers had trouble accessing these foods, worsened by limited
online delivery slots [11, 19, 21]. To address panic-buying, the government relaxed competi-
tion laws and supermarket driver limits, while supermarkets rationed products and introduced
special hours for key workers [11–13, 22]. The UK parliamentary Environment Food and
Rural Affairs Committee deemed the government's food system resilience plan for Covid inad-
equate, citing poor communication and lack of public reassurance [10, 12, 13]. The govern-
ment argued retailers were best suited to handle supply chain communication but faced
criticism for shifting responsibility to them [11, 12, 14].

### Panic-buying and social inequity

Panic-buying contributed to food system inequity, with higher socio-economic groups able to
stockpile more readily [16, 23–25], while those on Universal Credit couldn't shop for the best
prices due to limited availability [17]. The vulnerable, marginalised, and disabled lacked food
access [11, 13, 23–26]. Panic-buying also reduced the stock available at food banks [27]—a
food security lifeline, which almost doubled in demand during the pandemic [7, 10, 28].

### The media's role in shaping public behaviour

While panic-buying is often understood as a reaction to uncertainty, the role of the media in
potentially amplifying these behaviours is important. The media is a primary source of infor-
mation during crises, informing public perception and influencing behaviour through framing
and reporting choices. Evidence suggests that increased exposure to information about Covid
directly increased panic-buying [29]. At the beginning of the pandemic, 24% of the UK popu-
lation accessed Covid news 20+ times a day, 99% at least once a day and 43% used newspapers
as sources of information [30]. Ting et al. [31] refer to this information overload as an

'infodemic' and argue that it impinges consumers' ability to establish trustworthy information sources.

## Literature review

### Research gaps in panic-buying studies

Panic-buying is episodic, rare, and often specific to certain locations and disaster circumstances, limiting research potential, especially for empirical studies and lived-experience research [32–35]. Although academic literature on panic-buying peaked during Covid, with 85.7% of articles published between 2020 and 2021 [36], it remains under-researched [26, 32, 37]. Specifically, studies on the relationship between panic-buying and media are sparse [26, 37–42], with no media content analyses of Covid panic-buying in the UK press. However, there is a richer body of research on Covid panic-buying and social media [21, 28, 43–52].

### The causes of panic-buying

Research suggests four key factors which drive panic-buying: perceived scarcity, anticipated regret, fear and the need for coping mechanisms [32]. A study by Sherman et al. [29] also found that heightened anxiety contributed to increased panic-buying behaviours. Panic-buying can also be seen as a rational response to unusual circumstances where stocking up for longer periods is a means of survival [26, 43, 53–56]. Sherman et al. [29] argue that labelling the behaviour as panic-buying may be a misrepresentation, as it fails to account for both anxiety-driven actions and rational preparedness, both of which can explain consumer stockpiling during a pandemic.

### Sensationalism and panic-buying

The literature suggests that the media can play a role in causing panic-buying by using sensationalised language and images of empty shelves, which create scarcity cues [21, 26, 37, 53, 55, 57, 58]. According to Rajkumar [38] poor quality reporting in Covid likely exacerbated panic-buying. One global media content analysis showed 67.3% of the news articles during the pandemic had images of empty shelves and over half reported specific food items in short supply [37]. Ntontis et al. [33] argue that the media's description of specific foods increased demand of said goods. Media headlines during Covid were often sensationalised [38, 39] and the media may have magnified the potential risks of panic-buying, encouraging people to stockpile.

Prior to the first UK lockdown, English newspapers reported that people had started 'hoarding for a coronavirus outbreak' [59]. Examples of sensationalised language were words such as 'rampage' and 'riot'. Newspapers featured images of empty shelves and reported food shortages. Panic-buying had also been reported by UK media during Brexit [60], the 2021 lorry driver crisis [61], the 2022/2023 Ukraine war [62], and during the 2023 fruit and vegetable shortages [63].

### Theoretical frameworks and media influence

This study draws on agenda-setting theory and framing theory to examine how the media influenced public behaviour during the Covid pandemic. Agenda-setting theory explains how news media directs public attention by emphasising particular issues and how they are presented [64, 65]. Framing theory complements this by focusing on the way media outlets influence public perceptions through their choice of language, imagery, and tone when reporting [66]. Both theories are highly relevant during crises when the media play a critical role in providing information and directing public opinion [64, 65, 67]. Additionally, this study draws on

the view that photography and imagery used in media reporting can steer consumer behaviour by shaping visual perceptions of scarcity or abundance [68, 69].

The literature indicated that the media during Covid often blamed consumers, industry and government for panic-buying. Individuals were called greedy and selfish, and accused of depriving others of food [37, 39]. In contrast, supermarkets were portrayed as heroes and were the most quoted stakeholders [39].

By framing panic-buying as a crisis through images of empty shelves and sensationalist headlines, the media may have reinforced public fears and driven behaviours that exacerbated the problem. By applying these theories, the study seeks to understand the media's role in shaping the public's response and influencing policy debates around food system resilience.

## Research aims and contribution

Arafat et al. [37] argue that in using terms like 'panic-buying' and images of empty shelves, the media engaged in 'harmful reporting' about the availability of goods. Ntontis et al. [33] confirm these findings in their study of UK stockpilers, and Coleman et al. [60] found that media framing during Brexit may have contributed to panic-buying.

The evidence calls for the implementation of media reporting standards, restrictions on media reporting and monitoring mechanisms [37, 38, 70–73]. Dulam et al.'s [73] modelling study found that reducing media coverage of panic-buying by 50% led to a decrease in demand and increased product availability. The literature emphasises that images of empty shelves (often seen as a symbol of panic-buying) and sensationalist language should be minimised to prevent reinforcing panic-buying behaviour [39, 70, 74].

This study aims to analyse the agendas set by print media that informed public opinion, by analysing text and visual data. The media can set the agenda by making one element of the news more prominent than another or by framing news items in a certain way [65]. Visuals, including photos, symbols, and imagery, influence how the public and policymakers interpret the significance of these events [75–77]. The findings suggest that media analysis can be a valuable tool for understanding food system issues by highlighting: the media's role in assigning blame, revealing which items were perceived as scarce, showing patterns in consumer behaviour and offering real-time commentary on both causes and effects of panic-buying. The study contributes to agenda-setting theory and holds important implications for food policy, particularly in enhancing resilience and communication strategies for future food system disruptions.

The research also addresses the gap identified in the literature: there has been little analysis of how the British press portrayed panic-buying during Covid. A better understanding of media portrayals is essential for improving future crisis response, as highlighted by the Institute for Fiscal Studies [19].

## Research questions (RQ)

1. **RQ1**: What were the trends in UK media reporting on panic-buying from March to July 2020, and what imagery was used?
   This question examines how often panic-buying was reported and explores the visual cues, such as empty-shelf images, that may have reinforced panic-buying behaviours [26, 53, 57].

2. **RQ2**: What were the dominant themes and narratives, and how was the issue framed?
   This question looks into the key themes, narratives, and sources used in media reports, evaluating whether the coverage was balanced or sensationalised, and how it may have influenced public and policy views [55, 58].

3. **RQ3**: What insights from media portrayals of panic-buying can inform future policy planning for food system resilience?

This question seeks to understand how media coverage informs policymakers in planning for future crises, including resilience strategies, communication improvements, and the role of supermarkets in managing food supply shocks. By answering these questions, the study will provide important insights into the communication of food system crises.

## Methodology

This is a mixed-method media and visual content analysis, analysing articles published in the UK's most popular newspapers from March 2020—July 2020. Content analysis refers to a suite of methods used to infer meaning from text and visual data [78]. Visual content analysis, specifically, is a method of assessing how the media represent issues, people and events by systematically coding visuals to generate evidence [79, 80]. This research is grounded in agenda-setting theory [75] and seeks to establish how food policy issues in times of crisis are intensified by the media. The content analysis built on methods developed by Buckton et al., [81] and Wells and Caraher [65], which analyse media coverage of food systems issues -systematically documenting newspaper coverage to determine frames and themes [65]. When analysing articles, the lead author examined two types of content—manifest and latent [65]. First, the visible or surface-level details, known as 'manifest content'–this includes easily identifiable features, such as the article's title, publication date, number of words, and the number of images. Second, examining the underlying themes and messages is called 'latent content'. This refers to the deeper meanings, patterns, or narratives in the articles and images, which require more detailed analysis to uncover.

## Sampling

Despite declining newspaper readership, the national press still reaches a substantial audience, with a daily circulation of 10.4 million and 49% of adults claiming to read an online or print newspaper daily [82]. This makes newspapers a credible and relevant source for media content analysis, particularly during a period of public crisis. The lead author selected six titles and their Sunday equivalents from the highest-circulating newspapers: *The Daily Mirror*, *The Sun*, *The Independent*, *The Guardian*, *The Metro* and *The Daily Mail* [83]–two news brands from each genre. These titles were chosen to reflect a broad range of media genres, political stances, and readership demographics. The study included a mix of popular, mid-market, and quality titles, ensuring a comprehensive analysis across the spectrum of British newspapers [84].

Crucially, the political stance of these newspapers is to direct their framing of panic-buying. For example, *The Sun* and *The Daily Mail* are traditionally associated with right-leaning perspectives, whereas *The Guardian* and *The Independent* are left-leaning. Such political biases can affect how the causes and consequences of panic-buying are portrayed—whether it is framed as a failure of government policy, consumer behaviour, or supply chain issues. By including a range of political stances, the analysis aims to capture how ideological perspectives shape media coverage of the pandemic.

Furthermore, titles were selected to capture a wide range of readership demographics. *The Sun* and *The Daily Mirror* primarily reach C2DE audiences, while *The Daily Mail* appeals to a higher percentage of 65+ ABC1 audiences. *The Guardian* and *The Independent* attract a larger proportion of younger readers, particularly 16–24-year-olds [85]. These variations in readership guide how newspapers frame and present issues such as panic-buying, with each outlet tailoring its content to resonate with the unique concerns and interests of its audience.

Most newspapers operate as multi-platform news brands, offering both print and online editions. To enhance the scope of this study, both print and online titles were included in the analysis. This dual approach not only broadened the reach of the study but also ensured that the portrayal of panic-buying was captured across different formats. Given that online readership tends to be younger compared to traditional print [79], analysing both platforms provided a more nuanced view of how the issue might be perceived by different age groups [82].

A time-period of March 1 2020 to July 31, 2020 was chosen to incorporate the pre-lockdown period, when stockpiling began, and the easing of restrictions in August 2020 [2]. The geographical location was restricted to England. A list of articles used in analysis is available in S1 File.

## Search strategy

The lead author used a methodical process to reach a final sample of articles and images. This involved searching for articles, removing irrelevant articles, and reading the remaining articles in full and assessing against an inclusion/exclusion criteria [Fig 1]. The lead author used Nexis, a news and business research database to access data and the #atleast3 search limiter for 'panic-buying' or 'stockpiling' AND food. Headlines were read and irrelevant articles excluded, with remaining articles exported for further analysis. Articles were excluded if they were 1) advertorials, comments, duplicates or letters 2) not related to panic-buying or stockpiling and food or related to non-food goods or Brexit and 3) not in England.

The search strategy resulted in a final sample of n = 209 articles. For the visual content analysis, the lead author required the media articles in situ to view the images and place the data into relevant contexts [86]. However, one significant limitation of Nexis is that it does not provide access to the visual content, such as photographs or images, that accompanies the text in articles. This lack of access posed a challenge to conducting a comprehensive visual analysis. To address this limitation, the lead author manually searched for the corresponding online articles using Google, entering headlines and partial content to locate the original sources. This method allowed for 85% of the articles (n = 179) to be replicated online. Of these, 125 articles (70% of the sample) contained photography relevant to panic-buying. While this approach

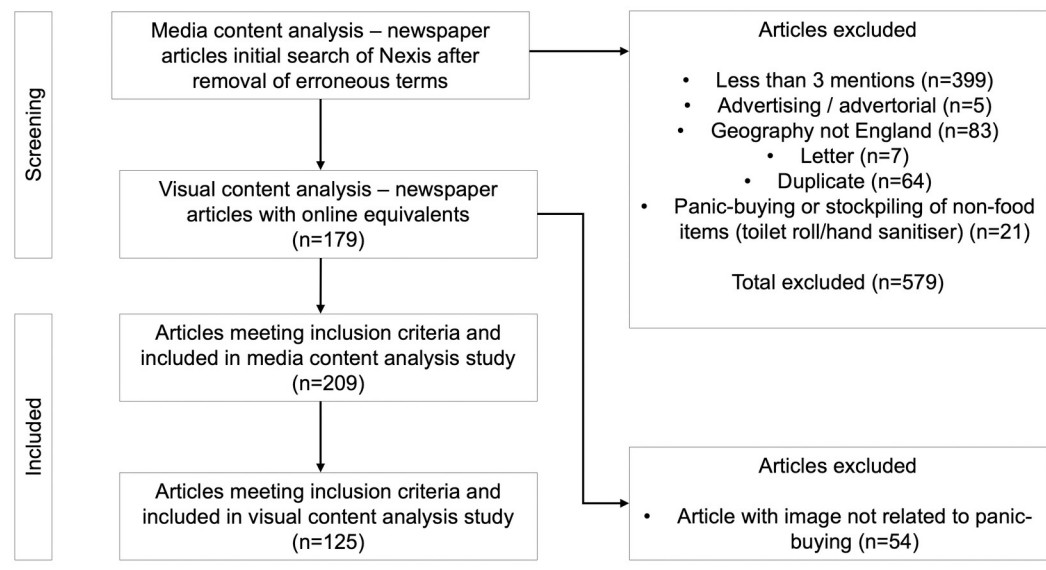

**Fig 1. Flowchart of article selection.**

was effective for a majority of the sample, it's important to note that the inability to retrieve the full visual content directly through Nexis may have limited the comprehensiveness of the visual data collection.

## Development of the coding framework

A coding framework was developed to allow for the systematic recording of data and imagery within the articles (S2 File), closely following the frameworks developed by Buckton et al. [81] and Wells and Caraher [65]. Manifest data in this study included the newspaper title, publication date, stakeholder voices, headline and image category. This allowed for the development of temporal and reporting trends analysis.

## Analytical framework overview

To provide clarity and transparency, the research methodology was organised into three key stages, illustrated in Table 1. This tiered approach integrated content and visual analysis, systematically coding manifest and latent data, and linked these findings to the research objectives.

The lead author also tracked sentiment and sensationalism using the framework in Table 2. This analysis of 'loaded words' [87] provides an understanding of how language and texts influence and sustain social meanings and knowledge structures [88].

For the visual content analysis, the author reviewed the body of images and coded these into five categories. These categories included, for example: images of empty shelves (which signal scarcity or panic), images of long queues (which suggest public anxiety) or photographs of stockpiled goods (which visually represent the behaviours being reported on). The systematic coding of images allowed the research to explore how visual cues may have been employed to influence public perception. For example, repeated imagery of empty shelves or crowded supermarkets created scarcity cues that may have influenced readers' decisions to panic-buy.

**Table 1. Analytical framework overview.**

| Stage | Focus | Method | Data collected | Connection to results |
|---|---|---|---|---|
| 1 | Content analysis | Coding of manifest content (e.g., publication date, stakeholder voices, headlines). | Visible data such as title, date, word count, image count. | Establishes reporting trends and temporal patterns. |
| 2 | Latent content analysis | Coding of themes, frames, and narratives using latent content (e.g., tone, language, underlying messages). | Thematic codes such as socio-economic impacts, food access, supermarkets. | Links media narratives to public perception and agenda-setting. |
| 3 | Visual content analysis | Systematic coding of images to assess contribution to panic-buying framing (e.g., empty shelves, long queues). | Visual cues such as scarcity, panic, and public behaviour. | Reveals visual elements that may shape public responses. |

**Table 2. Framework for defining sentiment and sensationalism.**

| Tone | Words/sentiments |
|---|---|
| Positive | Panic-buying as altruism/protecting loved ones. |
| Negative | Panic-buyers described as selfish/self-serving and impacts as catastrophic, evidence of shortages, empty shelves and riots. |
| Neutral | Panic-buying is described as fact or using balanced tone. |
| Mixed | Mixture of positive and negative tones. |
| Sensationalism | Use of words such as 'wave', 'deluge', 'riots', dramatisation of impacts–'giant rats', 'chaos', 'empty shelves', 'warning'. Implied threat. |

This aligns with the study's core aim to understand how media visuals contributed to agenda-setting and the public's perception of food access during the crisis.

For the latent content, the author drew on Entman's [66] definition of framing, where narratives are constructed to promote the interpretation of an issue in a particular way. The articles were examined in detail to identify frames and agendas. The coding process revealed six frames: prominence of supermarkets; food supply/access; food policy; individual behaviour; socio-economic impacts of panic-buying and panic-buying drivers. Outside of these, certain tensions and contradictions emerged, such as media outlets advising customers to stockpile [59] and subsequently criticising those who did [89], these were also logged onto the coding frame.

The lead author tested the coding framework on 15 articles to validate the coding categories. In the final coding phase, the lead author carefully read each article, systematically recording both manifest and latent data in Excel. For each thematic code, the author noted whether the article explicitly contained content related to that theme. For example, *The Sun*, 4.03.2020, which specified 'supermarkets have plans to ration food by working with suppliers to cut back on variety—focusing instead on their supplies of staple products,' was coded as mentioning food supply, supermarkets and supermarket food policy. Both quantitative and qualitative research methods were employed, with statistical analysis performed manually in Excel. This dual approach—quantitatively measuring reporting trends alongside qualitative exploration of narratives, themes, and frames—provided a comprehensive interpretation of agenda-setting during Covid.

## Results

### Reporting trends

The database search returned n = 788 articles. Following the inclusion/exclusion criteria the manifest data shows a total of n = 209 relevant articles were printed between March 1 –July 31, 2020. Table 3 shows the breakdown of articles across newspaper titles with their genre and political stance. The popular press comprised over half the articles and coverage was led by left-wing press (46%). *The Daily Mirror* reported most frequently in this period, with 35% of articles, followed by *The Daily Mail* (19%) and *The Sun* (19%).

Although the study cannot definitively explain why *The Daily Mirror* led in coverage, existing literature provides some hypotheses. Tabloids like *The Daily Mirror* focus on sensational, emotionally charged stories that resonate with their audience [90]. Panic-buying, a significant

**Table 3. Article breakdown across publications.**

| Newspaper | Genre | Political Stance | No of articles | % of articles |
|---|---|---|---|---|
| *The Daily Mirror* | Popular | Left-wing | 73 | 35% |
| *The Daily Mail* | Mid-Market | Right-wing | 40 | 19% |
| *The Sun* | Popular | Right-wing | 39 | 19% |
| *The Independent* | Quality | Centre | 25 | 12% |
| *The Guardian* | Quality | Left-wing | 24 | 11% |
| *The Metro* | Mid-Market | Neutral | 6 | 3% |
| *The Mail on Sunday* | Mid-Market | Right-wing | 1 | 0% |
| *The Sunday Sun* | Popular | Right-wing | 1 | 0% |
| *The Sunday Mirror* | Popular | Left-wing | 0 | 0% |
| *The Observer* | Quality | Left-wing | 0 | 0% |
| | | Total | 209 | 100% |

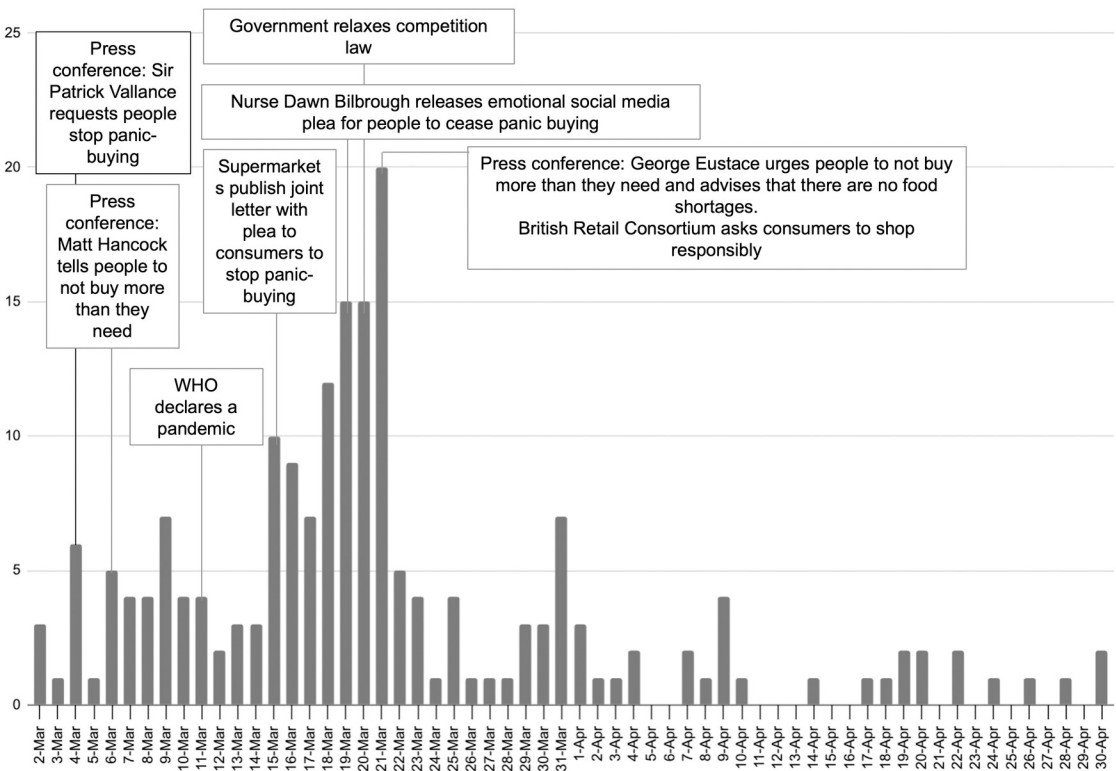

**Fig 2. Number of articles in March and April, annotated with potentially triggering events.**

public concern during the pandemic, aligns with this focus. *The Daily Mirror*'s readership, primarily C2DE social groups [82, 85], may have been more impacted by shortages, prompting the newspaper to prioritise this issue to reflect their concerns. Additionally, as a left-leaning paper, *The Daily Mirror* often critiques Conservative policies [91], and panic-buying provided a lens for highlighting governmental shortcomings. These factors may have contributed to its extensive coverage, though these remain informed hypotheses.

96% of the articles were published in March/April; March accounted for 82%. Given the concentration in March/April, Fig 2 shows the reporting peaks for this period only and identifies political, news, or global announcements. Between March 18–21, coverage peaked at 62 articles. This intense focus could reflect intra-media agenda setting, where media outlets not only report on the same issues but influence each other's focus and framing of topics, leading to a consistent narrative across various sources. These articles focused on supermarket rationing policies and human-interest stories about the elderly or vulnerable. A social media video by nurse Dawn Bilbrough [92] pleading with shoppers to stop panic-buying may have driven coverage, alongside announcements by government figures Sir Patrick Vallance (former Government Chief Scientific Advisor), Matt Hancock (former Health Secretary) and George Eustace (former Secretary of State for Department for Environment, Food and Rural Affairs).

## Imagery trends

There were certain images which became symbols of the pandemic, such as empty shelves and queues. Analysis in Fig 3 shows empty shelves were in 64% of the sample and queues in 40%.

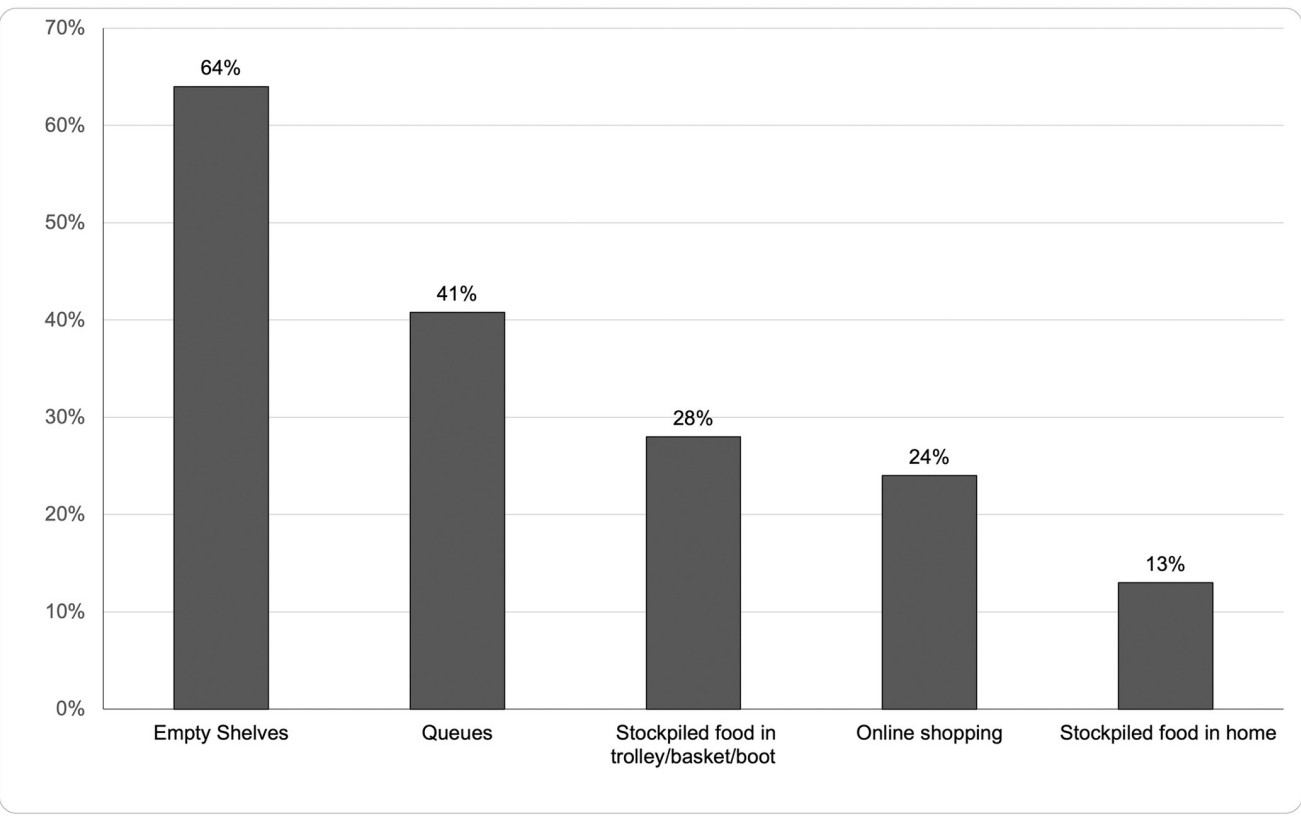

**Fig 3. Percentage of articles in sample which showed image type.**

Online shopping (19%) varied from images of unavailable delivery slots on supermarket websites to images of social media users complaining about a lack of slots. Newspapers often used the public's social media posts of empty shelves to illustrate specific food shortages. 91% of empty shelf imagery was shown by the popular and mid-market genres–there was a notable gap in empty shelf imagery in the quality press. Wright argues that visual imagery conveys information and can increase media impact [93]. The frequent use of empty shelf imagery is an example of how visuals can reinforce the public's sense of urgency, potentially driving consumer behaviour. This supports the agenda-setting theory by showing how media can direct attention to specific issues.

### Sentiment analysis and sensationalism

Sentiment and sensationalism were coded according to Table 4. Sentiment analysis showed most articles were negative (68%). Negative articles were driven by four main areas: panic-buyers described as self-serving; the chaotic situation; impacts on the vulnerable and inadequate government response. Positive stories (2%) recounted supermarket efforts to help the elderly or consumer confessions about not panic-buying. 20% combined both negative and positive slants and 9% were neutral.

Sensationalised language may imply impending threat, dramatise situations or vilify people. Table 4 gives an indication of the sensationalist language used in headlines. This provides insights into how language is used to produce meaning [88]. There was a high degree of sensationalism across the articles [89%], evenly spread across all genres. Quality press, such as *The*

**Table 4. Examples of sensationalised language.**

| Sensationalised language | Source |
|---|---|
| 'A major outbreak of the virus could result in " panic buying, empty shelves and food riots"'. | (*The Guardian*, 2.03.2020) |
| 'Supermarkets brace for wave of stockpilers as UK coronavirus fears escalate'. | (*The Daily Mirror*, 2.03.2020) |
| 'DON'T PANIC? SHOPPERS CLEAR STORES'. | (*The Daily Mail*, 4.03.2020) |
| 'RAT RACE Rats invading homes and turning into 'cannibals' due to coronavirus starvation after restaurants shut.' | (*The Sun*, 19.04.2020) |

*Independent*, had sensationalism scores similar to popular titles, such as *The Daily Mirror* (84% and 88%, respectively). This could be indicative of 'tabloidization', where mid-market and quality newspapers compromise on quality output to increase readership [94]. The high level of sensationalism found in this study aligns with previous research on how emotionally charged reporting may have affected public panic and stockpiling behaviours, further contributing to the media's agenda-setting role during crises [21, 26, 37–39, 53, 57, 58, 60].

## Stakeholder voices

The author analysed stakeholder voices across industry (supermarkets and retail organisations), political (government representatives or organisations), experts (food systems and logistics specialists), and consumers. The most frequently cited source was industry, appearing in 62% of the articles. Supermarkets aimed to reassure the public about the absence of food shortages and discouraged stockpiling.

Supermarkets often dominated the headlines, oscillating between arbiters of shopping rules ('VERY LITTLE HELPS STORES LIMIT STOCKS AMID BUG FEARS–Tesco rations pasta & veg to curb panic buys'—*The Sun*, 9.3.2020), and heroes who kept the nation fed ('UK supermarkets draw up plan to 'feed the nation' as coronavirus spreads' -*The Guardian*, 2.03.2020). A joint letter from supermarkets via the British Retail Consortium [5] urging shoppers to be considerate also dominated headlines, which may suggest that retailers, rather than the government, were leading the response to panic-buying.

Political voices appeared in 27% of articles, with government sources stating that panic-buying was unnecessary and supply chains were resilient. Consumer voices featured in over a third of articles, with detailed accounts of panic-buying behaviours by individuals ('filled his boot with 15 kg of penne, 48 bags of crisps, 16 tins of beans, and litres of Dettol,'–*The Daily Mail*, 2.03.2020) and distinctions between panic-buying and cautious preparation ('It's not panic-buying, I think people are just being cautious and buying stuff for storage,'–*The Guardian*, 2.03.2020). Evidence suggests that social media was a significant platform for sharing panic-buying behaviours [45], and this was reflected in this study, with many consumer voices presented through social media extracts and reprints.

Experts appeared in 30% of articles, often quoting Bruno Monteyne, a City of London analyst and former Tesco supply chain director. His quotes were used to create sensational headlines, such as 'Coronavirus: Supermarkets preparing for 'food riots' as panic-buying Brits strip shelves' (The Daily Mirror, 6.03.2020).

The prominence of supermarkets as the primary source of reassurance suggests that media narratives framed them as central to resolving the crisis, highlighting the media's role in shaping public understanding of key institutions during the pandemic.

## Thematic analysis

This section describes the main themes and sub-themes in newspaper coverage, analysed through the lens of agenda-setting theory [75], which posits that media story selection shapes the public agenda [95]. It examines how media frames the news agenda through specific narratives [65, 77]. It also offers analysis of the visual imagery accompanying reporting.

## Overall findings

Six main themes were identified in the newspaper coverage. Supermarkets were mentioned in 88% of articles, followed by food supply/access (79%), food policy (74%), individual behaviour (72%), socio-economic impacts of panic-buying (70%), and panic-buying drivers (41%). Table 5 outlines these themes and sub-themes, with example headlines illustrating their presentation.

## Sub-themes

Table 6 shows which sub-themes were mentioned most. Supermarkets, for example, were mentioned in 88% of articles and 67% of articles mentioned supermarkets' food policy.

Fig 4 shows the breakdown of all sub-theme mentions.

**Theme 1: Prominence of the supermarket (88%).** Supermarkets dominated the panic-buying narrative (in 88% of articles), which aligns with prior research [39]. Supermarket empty shelves prevailed in the accompanying visual imagery. In contrast to the narrative of supermarkets unable to cope, newspapers also hailed supermarkets as saviours, frequently using the wartime phrase 'feeding the nation'. They were portrayed as policy leaders, driving solutions to panic-buying, and leading on communications–a position concreted by the government:

> *The retailers reassured me they have well-established contingency plans and are taking all the necessary steps to ensure consumers have the food and supplies they need.* (George Eustace).
>
> (*The Daily Mirror*, 6.03.2020)

There was a notable absence of any alternative food systems in the narrative–with only one article mentioning corner shops (*The Guardian*, 22.04.2020). The media's emphasis on supermarkets as key players in managing the crisis is consistent with agenda-setting theory, portraying these institutions as the primary solution to food supply challenges.

**Theme 2: Food supply/access (in 79% of articles).** Food shortages were mentioned in 20% of articles, while the food supply chain appeared in 35%, fluctuating between being portrayed as resilient and strained. There was a contrast between claims of no shortages and reports of specific items being scarce. Empty shelves were highlighted in 58% of articles, increased demand in 38%, and supermarkets struggling to meet demand in 19%. Additionally, 23% mentioned a lack of delivery slots, with 19% showing images of online shopping. People were panic-buying in-person and digitally. By emphasising food supply disruptions, the media may have heightened public anxiety, and reinforced panic-buying behaviours.

**4.6.5. Theme 3: Food policy (in 74% of articles).** Supermarket food policies were mentioned in 67% of articles, with a focus on rationing. The articles frequently outlined which items were being rationed and many provided rationing rules by store, which was a useful public information service. Similar reporting provided information on which delivery slots were available by store. Other policies referenced were shopping hours for the vulnerable and elderly, operating hours and streamlining product ranges. The policies were aimed at either

**Table 5. Panic-buying main themes and sub-themes.**

| Main theme | Sub-theme | Headline examples |
|---|---|---|
| **Prominence of supermarkets** | Supermarkets 'feeding the nation' | 'UK supermarkets draw up plan to 'feed the nation' as coronavirus spreads; [. . .]', (*The Guardian*, 2.03.2020). |
| | Supermarkets rising to the challenge | 'Supermarkets ready for a new week of rising to the virus's challenge; Supermarkets have won praise for their response to the crisis (. . .), (*The Guardian*, 29.03.2020). |
| | Supermarkets leading on communications | 'Coronavirus: Tesco top boss tells shoppers not to panic as 'there's plenty of food', (*The Daily Mirror*, 12.03.2020). |
| **Food supply/access** | Supply chain impacts | 'Ocado warns of delivery delays as Brits start hoarding for a coronavirus outbreak', (*The Daily Mail*, 2.03.2020). |
| | Specific food shortages | 'UK supermarkets ration toilet paper to prevent stockpiling. Other vanishing items include dried pasta, tinned vegetables, medications and hand gel', (*The Guardian*, 8.03.2020). |
| | Empty shelves | 'IT'S SHELF-ISOLATION; PANIC-BUYING HITTING SUPERMARKETS struggle to restock amid frenzy Chaos as Health Sec's contradicted', (*The Sun*, 7.03.2020). |
| | High demand/supermarkets unable to cope / Lack of online delivery slots | 'SELL OUT Supermarket delivery slots sell out over coronavirus stockpiling and shoppers say, 'it's worse than Christmas'', (*The Sun*, 8.3.2020). |
| **Food policy** | Supermarket policies to reduce panic-buying | 'Virus CHAOS Rationing hits supermarkets for the first time since WWII', (*The Sun*, 6.03.2020). |
| | Government policies aimed at easing supply chain issues | 'PANIC PLANS Government could relax delivery time rules to help supermarkets deal with coronavirus pressure', (*The Sun*, 7.03.2020). |
| **Individual behaviour** | Description of the stockpiled food | 'Man's heart breaks overhearing elderly shopper ask for eggs (. . .)', (*The Daily Mirror*, 19.03.2020). |
| | Queues | 'QUEUE STRETCHES THIRD OF A MILE AS BUYERS STOCK UP', (*The Metro*, 20.02.2020). |
| | Demonisation of the stockpiler/panic-buyer | ''BLOODY IDIOTS' Furious husband slams 'horrible' coronavirus panic buyers after Tesco worker wife left in tears after 'worst day ever'', (*The Sun*, 20.03.2020). |
| | Abuse and rioting | 'Supermarket staff report rise in abuse amid coronavirus panic-buying of toilet rolls, cleaning products, pasta and tinned food despite calls for calm from shoppers', (*The Daily Mail*, 16.03.2020). |
| **Socio-economic impacts** | Disabled, elderly or vulnerable unable to secure food | 'Disabled people cut off from vital supplies due to panic buying (. . .), (*The Guardian*, 18.03.2020). |
| | Impact on food banks | 'Food banks run out of milk and other staples as shoppers panic-buy (. . .), (*The Guardian*, 10.03.2020). |
| | Food waste increase | 'Giant rats 'could invade streets and homes' as COVID-19 stockpilers dump spoiled food', (*The Daily Mirror*, 30.03.2020). |
| | Increased consumer spending | 'SO SHELF-ISH Coronavirus panic-buying shoppers spent £60million extra in first week of March', (*The Sun*, 20.03.2020). |
| | Supermarket profiteering | 'AND EVEN IF STOCKPILING FIZZLES OUT, STORE CHIEFS EXPECT PROFITS SURGE TO RUN ON FOR MONTHS', (*The Daily Mail*, 22.03.2020). |
| **Panic-buying drivers** | Fear and anxiety / stockpiling as a safety net | ''It's a safety net': across the UK people stock up amid coronavirus fears; Guardian readers say they are concerned about being forced to self-isolate at home', (*The Guardian*, 2.03.2020). |
| | Lack of faith in government | 'Supermarkets blast Matt Hancock's 'fanciful, bogus and misleading' claims about mass food deliveries to the sick,' (*The Daily Mail*, 7.03.2020). |

**Table 6. Percentage of articles that include the most mentioned sub-theme within each main theme.**

| Main theme | Sub-theme | Percentage of articles |
|---|---|---|
| Prominence of supermarket | Mention of supermarket | 88% |
| Food Policy | Supermarket food policy | 67% |
| Individual behaviour | Description of food panic-bought | 61% |
| Food supply/access | Empty shelves | 58% |
| Socio-economic impacts | Disabled, vulnerable or elderly | 56% |

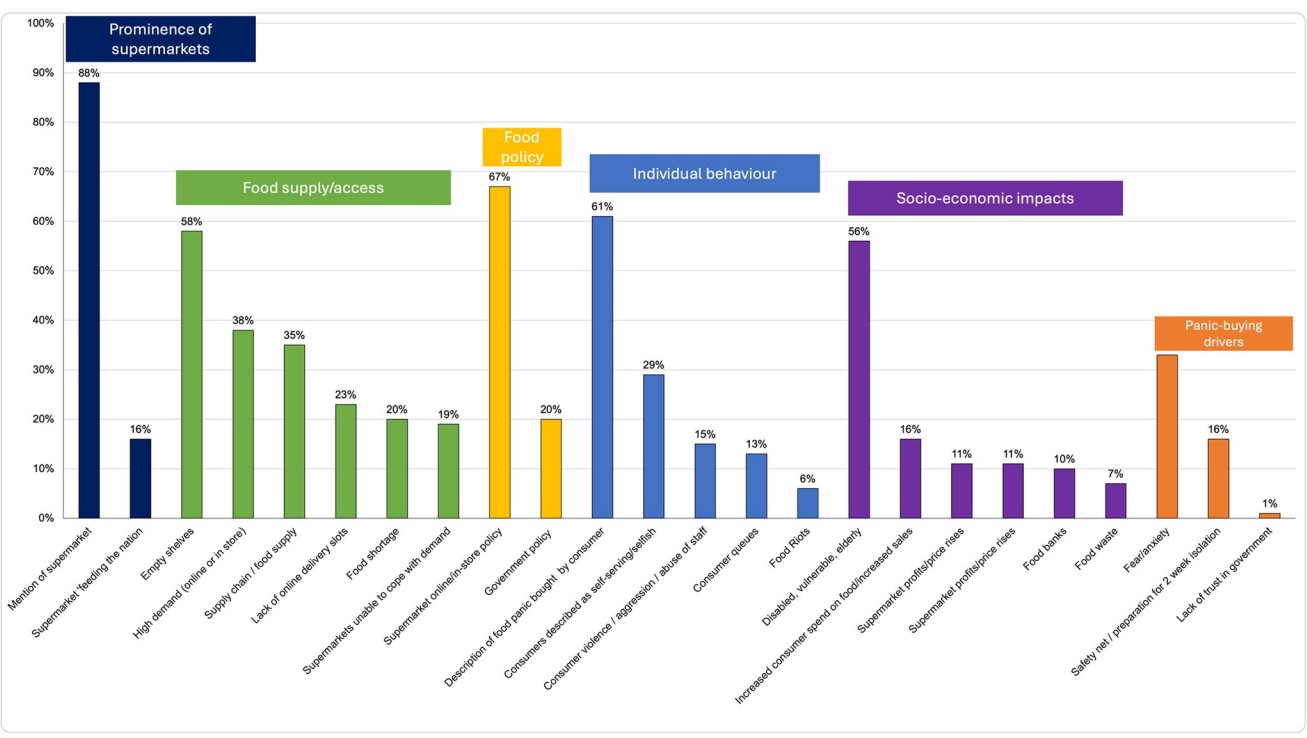

**Fig 4. Percentage of articles which mention sub-theme.**

reducing panic-buying at point of purchase ('a bid to stop shoppers from stockpiling'-*The Daily Mail*, 1.04.2020) or easing supply chain issues. The articles also mentioned online policies such as supermarkets adjusting delivery slots. Only 20% of articles mentioned government policies on panic-buying. These included measures such as reducing driver hour restrictions and relaxing competition law. The media's coverage of food policy decisions may have steered public expectations of institutional responsibility, demonstrating the agenda-setting function of the press in shaping trust during times of crisis.

**Theme 4: Individual behaviour (in 72% of articles).** The foods purchased by individuals were often listed (61%)–'she topped up on porridge, lentils, rice, pasta and tinned tomatoes' (The Guardian, 2.03.2020). There were frequent mentions of shoppers' locations. In addition to shelf-stable goods, frozen food was portrayed as a stockpiling staple.

The selfishness of individual behaviour was frequently called out by newspapers (29%). Stockpilers were portrayed as selfish and demonised–'a 'grim' example of the selfishness of humanity' (The Daily Mail, 5.03.2020), 'the locusts had been' (The Guardian, 3.04.2020). There was similar negative sentiment to people queuing (mentioned in 13% of the articles). Queues were consistently described as 'huge' and words such as 'influx', 'wave' and 'surge' were used. Related to this, aggression and violence were mentioned in 15% of articles, with 6% of articles alluding to potential riots. Visual content analysis also showed that queues were shown in 51% of the sample. Panic-buyers were also consistently criticised for inconsiderately consuming food that the vulnerable needed. The negative portrayal of consumer behaviours during the crisis may have influenced social norms around panic-buying, reflecting how media framing may have impacted public judgments of others.

**Theme 5: Socio-economic impacts of panic-buying (in 70% of articles).** The dominant sub-theme in this category was the disabled, vulnerable, or elderly's access to food, as shown in

56% of all articles. This applies to access to both in-store food and online delivery slots. The inequity of panic-buying was outlined by newspapers, with those on lower socio-economic incomes not having available funds to panic-buy–'I have same fears and concerns (. . .), but obviously do not have the financial funds to bulk buy' (The Sun, 6.03.2020). 10% of the articles outlined how panic-buying impacted food banks. The media's focus on the socio-economic impacts of panic-buying may have influenced public and policy discourse around vulnerability.

**Theme 6: Panic-buying drivers (in 41% of articles).** Covid was a time of great uncertainty, with rapidly changing information and threat levels. The articles outlined how people were frightened about the possibility of being stuck without food for the 2-week isolation period and, therefore, panic-bought to create a food safety net, mentioned in 16% of the articles. Fear and anxiety featured in 33% of articles–'I'm anxious but maintain a veneer of calm and conceal it by prepping' (The Guardian, 2.03.2020). Lack of trust in government only made up 1% of mentions. The media's emphasis on specific panic-buying drivers, such as fear and anxiety, are indicative of repetitive narratives, these narratives may have reinforced consumer behaviour patterns.

## Visual imagery

64% of the sample included images of empty shelves, queues, and unavailable online delivery slots, creating a frightening perception of food shortages and increased demand. Emotive photos, particularly of the elderly in front of empty shelves, frequently appeared in newspapers and became prominent in the news agenda. For example:

' (Elderly man staring at empty Sainsbury's shelves shows panic-buying impact)'

(The Daily Mirror, 18.03.2020).

Additionally, social media posts detailing specific locations with empty shelves and showing images of food in people's boots or trolleys vividly illustrated the panic-buying phenomenon. The visual imagery acts as a concrete manifestation of the reported panic-buying. These may have reinforced public perceptions of scarcity, illustrating how visual framing by the media may have intensified consumer anxiety and driven behaviour, in line with agenda-setting theory.

## Tensions and contradictions

There were several tensions and contradictions within 60% of articles, which sat outside the themes but provided important insights. These were i) emotional pleas to cease panic-buying and encouragement of altruism (46%), ii) mentions of 'no shortage of food' and adequate supply (27%), iii) advising/discussing what consumers *should* stockpile (7%), and iv) mention of contradictory advice by the government (7%).

The plea for consumers to stop panic-buying (46%) came from multiple sources: retailers, government, newspapers and consumers. This tension firmly portrayed the panic-buying crisis as a fault of individuals and not a breakdown of the food system. 27% of the articles mentioned that there was no shortage of food or an adequate supply. These phrases are often attributed to retail or government sources. Images of empty shelves ran alongside articles which said 'adequate supply', in apparent contradiction.

This confusion may have been compounded by contradictory government advice (7% of articles). For example, newspapers reported on Public Health England's advice to stock up on two weeks' food supply in case of isolation, which sent 'mixed messages' (The Daily Mail,

2.03.2020). Furthermore, the Prime Minister advised people to visit the supermarket in person to make slots available for the vulnerable–which supermarkets disagreed with.

> (. . .) *The Prime Minister urged everyone to use food delivery services wherever possible, but the reality is that current demand vastly exceeds supply.*

> (*The Independent*, 17.03.2020)

The final contradiction relates to coverage in the first two weeks of March. There was an explicit paradox where newspapers either asked if people were stockpiling or advised them what to stockpile (in 7% of articles). The Daily Mail printed a 'What should you stockpile in a pandemic' advice guide, which listed cereals, grains, beans, tinned food and soft drinks (*The Daily Mail*, 13.03.2020). Similarly, *The Sun* printed a food stockpile checklist of 'things you might want to include in your coronavirus stockpile'. This included items such as pasta ('high in carbs and stores well'), rice ('can be used in loads of different meals') and baking goods ('to make bread'), (*The Sun*, 10.03.2020). In subsequent weeks, the newspapers would then vilify those who stockpiled, despite having advised them to do so. The products which the newspapers advised people to buy are also those which were in high demand during the pandemic.

The media's contradictory messaging—encouraging stockpiling while vilifying panic-buyers—highlights how inconsistent framing can create confusion and social tension, further emphasising the media's power in shaping public understanding.

## Discussion

Agenda-setting theory argues media reporting can shape how the public and governments interpret an issue and its importance [75]. Through analysis of mainstream media coverage of panic-buying, this study has explored how panic-buying was framed by the media and offered insights about what themes were dominant in reporting. It also offered a novel contribution by combining both analysis of text and accompanying visual imagery, which enhanced the understanding of how media may potentially drive public behaviour during crises.

### Reporting trends

82% of articles analysed were printed in March 2020, peaking between March 18–21. This coincided with a surge in consumer spending, with supermarkets experiencing their highest sales in the week before the first lockdown on March 23 [3]. Coverage spikes were linked to government announcements, a British Retail Consortium's open letter, and human-interest stories like nurse Dawn Bilbrough's plea against panic-buying. These peaks highlight how quickly news can steer media and public agendas, which is valuable information for policymakers. Uysal's study [43], also showed a correlation between media exposure and panic-buying, suggesting that understanding the types of content driving coverage can aid government communication planning. This research reinforces the need for policymakers to engage proactively with media outlets to promote preparedness rather than panic. Such efforts have the potential to foster food system resilience in times of crisis.

### Sensationalism and visual imagery in media coverage

While sensationalism is common in crisis reporting, the level observed in this study is particularly significant. 89% of the analysed articles used sensationalised language with terms like 'waves,' 'panic,' 'empty shelves,' and 'riots.' This aligns with Rajkumar's [38] and Phillips et al.'s findings [39], which noted high levels of sensationalism in media. Sensationalism,

coupled with visual cues like empty shelves (64% of articles) and queues (40%), and may have heightened public anxiety and stimulated panic-buying behaviours [36, 57, 60].

Arafat et al.'s study found that 67.3% of articles included empty shelves and Rajkumar found that 65% used imagery encouraging panic-buying [37, 38]. Notably, images of empty shelves often accompanied articles stating there was no food shortage, sending mixed messages. Consumers shared their panic-buying experiences in supermarkets, often with social media excerpts showing empty shelves, reinforcing perceptions of food shortages and encouraging panic-buying. Social media and print media portrayed food scarcity, with social media playing a key role in spreading panic-buying information [21, 45]. Combining sensational text and imagery may have compounded public uncertainty, further emphasising the need for responsible media reporting. Policymakers could consider introducing media guidelines to limit sensationalism and encourage accurate portrayals of food availability [60], which would help mitigate unnecessary public anxiety and panic-buying [36, 60].

## Supermarkets' dominance in coverage

Supermarkets appeared in 62% of articles, a finding consistent with Phillips et al. [39]. This finding supports Dixon and Banwell's theory of 'supermarketisation', where supermarkets hold significant power in the food system [96]. Supermarkets were framed mostly positively in 88% of articles, portrayed as 'feeding the nation', while negativity was often directed at individuals instead. Supermarket policies—such as rationing, streamlining product ranges, and distinct operating hours—were highlighted more than government policies (67% vs. 20%), with the government criticised for leaving the panic-buying response to retailers [11, 14].

Given the supermarkets' dominant role in media coverage and public perceptions during the pandemic, policymakers could strengthen collaboration frameworks with major retailers. This would enable consistent communication strategies to manage public expectations and avoid consumer panic in future crises.

## Themes and framing

Coverage predominantly blamed individuals for panic-buying, portraying them as immoral and supermarkets as heroes. Phillips et al. [39] noted that communities often blame specific groups while portraying others as victims or champions. This aligns with other studies on panic-buying media content [37, 39, 60]. In 46% of articles, the government, retailers, and consumers urged the public to stop panic-buying, blaming individuals for its impact on vulnerable populations. Common descriptors for stockpilers included 'selfish,' 'waves,' 'hoards,' and 'locusts.' Arafat and Kar support naming and shaming in their proposed media guidelines for panic-buying [97]. However, the authors argue that focusing on individuals diverts attention from the systemic nature of panic-buying, which requires active government involvement. Policymakers could use these insights to develop future public messaging, emphasising the need for systemic responses rather than individual blame.

**Fear and anxiety in coverage.** Over one-third of articles mentioned fear and anxiety, with 16% justifying stockpiling as necessary preparation for isolation. This raises the question of whether governments should advise consumers to stock up on food or maintain reserves themselves during crises. 61% of articles detailed what individuals were buying, usually long-life products like bread, eggs, and milk. Ntontis et al. [33] and Arafat et al. [72, 97] both argue that this level of specificity amplifies panic-buying behaviours. Policymakers could use this knowledge to adjust their communication strategies, ensuring that public messaging does not inadvertently drive demand for particular goods.

**Perceptions of food system resilience.**   A recurring theme was the portrayal of the food system as lacking resilience, with 35% of articles mentioning supply chain issues. Despite assurances from government and retailers that there were no food shortages, images of empty shelves persisted. Notably, evidence showed that 40% of consumers did not engage in panic-buying, suggesting the media's portrayal of widespread panic-buying may have been over-stated [98]. In the early stages of panic-buying, articles depicted supermarkets as struggling to meet demand, both in-store and online. The focus on online delivery slots (23% of articles and 19% of images) is important for policymakers, highlighting the food system's inability to han-dle fluctuating demand. This service is crucial during health crises when people must stay home. To address this, policymakers should prioritise clear and transparent communication with the public. Working closely with media outlets to disseminate accurate information about food supply can help reduce the risk of panic-buying in future crises.

**Vulnerable populations and food access.**   The findings emphasise the impact of panic-buying on the elderly, vulnerable, and disabled, who featured in over half the articles. These groups were disproportionately affected by food insecurity during the pandemic [99]. Policy-makers need to address how vulnerable populations can access essential goods during future crises. Developing targeted interventions, such as prioritising vulnerable groups for food deliv-ery services or ensuring access to essential goods, could help mitigate the adverse effects of panic-buying on marginalised communities.

**Contradictory messaging in media and policy.**   The study revealed conflicting narratives in the media, with articles about no food shortages appearing alongside images of empty shelves. This contradiction reflects poor communication and mixed messaging, as noted in Coleman et al.'s study [60]. The Environment Food and Rural Affairs committee criticised the government for over-relying on retailer communications, asserting that the government should lead messaging and legislation for shortages [10]. Additionally, early media reports advised con-sumers what to stockpile, only to later criticise panic-buying behaviour. Such contradictions highlight the need for clear and consistent messaging. Policymakers should work closely with media outlets to avoid contradictory advice and ensure public information is reliable and uni-fied. This would help minimise confusion and reduce panic-buying in times of crisis.

Although the media played a major role in shaping panic-buying behaviours, psychological responses like anxiety and government communication strategies also contributed. These find-ings underscore the importance of coordinated, clear messaging to avoid confusion in future crises.

## Implications

The media portrayal of panic-buying during Covid offers valuable insights for policymakers on food system resilience. Reporting on high-demand items like eggs, bread, and flour, and data from aggregators like Kantar, can help assess supply chain adaptability. This study highlighted that the media depicted the elderly, vulnerable, and disabled as highly affected by panic-buying, particularly impacting low-income individuals. Policymakers must ensure there are plans to feed the vulnerable during health crises; they could consider maintaining govern-ment reserves for emergencies and advising consumers to stockpile food [25, 36, 100, 101]. The government Prepare website [102] (launched in May 2024) offers some basic information on stocking up on non-perishable food such as tinned meat or fruit as 'emergency supplies'. However, it doesn't offer guidance for when people are advised to stay at home for long peri-ods of time for health crises, such as Covid.

The media narrative about supermarket policies during food system shocks and consumer reactions provides valuable insights for policymakers. The lack of focus on alternative food

**Table 7. Panic-buying media reporting framework for UK national press.**

| Principle |
| --- |
| Avoid excessive reporting on panic-buying: widespread coverage gives the impression that everyone is panic-buying, when research indicates that only a small percentage do. |
| Avoid sensationalised language to describe panic-buying. Avoid words such as 'surges', 'waves' or 'hoards'. |
| Take care when choosing visual images. Images of empty shelves and queues may contribute to feelings of scarcity and promote panic-buying. |
| Avoid contradictory narratives such as discussion of adequate food supply alongside images of empty shelves. |
| Reporting on specific food items in short supply or stockpiling may contribute to product demand. Avoid specifics, including locations, that might stimulate regional panic-buying episodes. |
| Avoid replicating viral social media posts, which adds to the perception that panic-buying is universal. |
| Avoid language that vilifies panic-buyers. This is divisive and may contribute to aggression. |
| Consider focusing on the negative impacts on others, such as the vulnerable, elderly or disabled, to inspire altruism. |
| Consider articulating coping methods, such as shopping locally [non-supermarket] or alternatives to high demand items (wholemeal versus white flour). |
| Work with the industry to communicate timely information on replenishment and items in a good supply rather than focusing on absence. |
| Work with the government to identify pathways to food for the vulnerable or food insecure [via local authority and government websites] and promote these prominently. |

systems like corner shops suggests a need for policymakers to consider their role in supporting food access during crises.

The evidence from this study may help inform government communications on panic-buying. A focus on how panic-buying limits food for the vulnerable and using a 'think of others' message instead of 'stop-panic buying' could be beneficial. George Eustace noted the government initially avoided discussing panic-buying to prevent it [12]. However, this analysis showed that in the absence of government communication, supermarkets dominated the narrative. Policymakers should review retailer-led communications and work with the media for clearer, consistent messaging on panic-buying. This study indicates the need for improved government messaging during food-system crises, highlighting inadequate communication during Covid. Collaboration with the media could ensure panic-buying reporting causes no harm and provides consistent, joined-up messaging.

Academic literature calls for media reporting guidelines to prevent mass, harmful coverage that may contribute to panic-buying [97]. While media standards exist for publishing accurate information [103], specific guidance for panic-buying could be beneficial. Policymakers should co-design these standards with the media to avoid compromising press freedom. The authors propose a new framework [Table 7] that draws on existing panic-buying media reporting guidance [38, 97] and respected NGO guidelines [104, 105].

The framework advises against excessive coverage to prevent reporting peaks. Unlike Arafat and Kar's [97] guidelines, which encourage shaming panic-buyers, this framework discourages it to avoid discord and civil unrest. It recommends using sensitive image choices, reducing sensationalism, and avoiding reports on specific foods and panic-buying locations. It also advocates for signposting resources for the food insecure, avoiding conflicting messages, promoting altruism, and focusing on food replenishment rather than shortages.

## Limitations

This study has several limitations, which may have affected the broader applicability of the findings. The sample was restricted to six of the highest-circulating newspapers, chosen to

represent a broad range of political perspectives, genres, and readership demographics. While this provides valuable insights into print and online news framing, the exclusion of social media, a platform known to significantly inform consumer behaviour during crises [49], limits the study's ability to capture the full scope of media framing of panic-buying. Social media is dynamic, real-time, and interactive, and its omission could mean certain drivers of panic-buying, particularly viral misinformation or user-generated content, were not captured. Including TV and radio, which were among the most widely used information sources during the pandemic [85], could have offered further insights into how audiovisual media contributed to shaping public perceptions and behaviours.

Additionally, this study focused only on the initial phase of panic-buying during Covid. Since subsequent episodes of panic-buying occurred during the pandemic, a longer-term study would provide more comprehensive insights into the evolving nature of media coverage over time.

Another limitation is the use of Nexis, which provides articles in plain text without context, such as advertisements, adjacent articles, or accompanying images, which may have impacted the depth of the analysis. While visual content analysis helped add context, the inability to view articles in their original layout, as seen by readers, may have limited the understanding of how these elements worked together. Additionally, images are subject to viewer interpretation and, without surrounding context, the full impact of these visual cues on the audience may be difficult to assess.

As one researcher undertook the coding, intercoder reliability was not possible, which is considered best practice [78]. However, the author took a systematic approach to the analysis of the articles using a structured coding frame and performed sample checks.

Although there is evidence to suggest a cause-and-effect relationship between media and the public behaviour [75], this study is limited to reviewing the content of media reporting and the author is unable to draw firm conclusions on media effects.

These limitations notwithstanding, this is a robust study of media text and visual analysis of panic-buying during Covid. The strengths of this study are that it was a mixed-method study, with visual content analysis overcoming some of the limitations of Nexis. Furthermore, a systematic analysis of a large number of articles was conducted using a structured coding frame. Analysing both manifest and latent content allowed for a richer analysis of the data. Lastly, a wide range of newspapers was used, incorporating the breadth of political spectrums, genres, and demographics.

## Conclusion

Panic-buying remains in media focus as the resilience of the UK's food system is frequently questioned [106]. Evidence indicates that food-system issues may lead to future civil unrest, with panic-buying playing a major role [107]. This study provides important insights into how the English press framed panic-buying during Covid and adds to agenda-setting and framing theory. It shows that sensationalised language and imagery of empty shelves, played a significant role in media coverage. This could potentially have amplified public anxiety and contributed to panic-buying behaviours. It highlighted key themes such as supermarket dominance, food supply, individual behaviour, and socio-economic impacts. It also identified the vital role supermarkets played in media narratives, often portrayed as 'feeding the nation,' while individuals were criticised for stockpiling.

These findings have several implications for policy. First, policymakers should consider collaborating closely with media outlets to ensure responsible reporting during crises. Guidelines limiting sensationalism and encouraging accurate depictions of food availability could help

mitigate public anxiety and prevent panic-buying. This is particularly important when evidence suggests that 40% of consumers did not panic-buy [98], indicating that the portrayal of widespread panic-buying may have been inaccurate.

Additionally, government communications should focus on clear, consistent messaging to avoid contradictory advice and maintain public trust. Policies ensuring food access for vulnerable populations during crises, such as prioritising delivery services or maintaining government reserves, are also essential.

Further media effects studies are needed to confirm whether media reporting exacerbates panic-buying, as suggested by the literature. This research fills a gap in UK media content analysis on panic-buying during Covid, contributing novel insights to the literature and building on agenda-setting and framing theory.

Future research could build on this study by expanding the media sample to include social media, television, and radio. Moreover, examining subsequent waves of panic-buying and their media coverage would provide further insights into how media framing evolves over time.

In sum, this study reinforces the importance of the media in shaping public behaviour during crises and offers actionable insights for policymakers to improve communication strategies and enhance food system resilience in future crises.

## Supporting information

**S1 File. List of newspaper articles used in analysis.**
(DOCX)

**S2 File. Framework for analysing newspaper article sample.**
(XLSX)

**S3 File. Excel spreadsheet of data and codes from articles.**
(XLSX)

## Acknowledgments

The authors would like to acknowledge and thank the Centre for Food Policy at City St George's, University of London.

## Author Contributions

**Conceptualization:** Dayna Brackley.

**Data curation:** Dayna Brackley.

**Formal analysis:** Dayna Brackley.

**Investigation:** Dayna Brackley.

**Methodology:** Dayna Brackley, Rebecca Wells.

**Supervision:** Rebecca Wells.

**Validation:** Rebecca Wells.

**Visualization:** Dayna Brackley.

**Writing – original draft:** Dayna Brackley.

**Writing – review & editing:** Dayna Brackley, Rebecca Wells.

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
