## [Decision Letter · Decision Letter 0]

29 Sep 2024

PONE-D-24-41107The portrayal of panic-buying and stockpiling in English newspapers during Covid, a mixed method content analysisPLOS ONE

Dear Dr. Brackley,

Thank you for submitting your manuscript to PLOS ONE. After careful consideration, we feel that it has merit but does not fully meet PLOS ONE’s publication criteria as it currently stands. Therefore, we invite you to submit a revised version of the manuscript that addresses the points raised during the review process.

We look forward to receiving your revised manuscript.

Kind regards,

Jiankun Gong

Academic Editor

PLOS ONE

Journal Requirements:

Additional Editor Comments:

Thanks for presenting this paper. Both reviewers gave a major revision of your paper. I personally read you paper and i found few significant papers should be included such as:

Sherman, C. E., Arthur, D., & Thomas, J. (2021). Panic buying or preparedness? The effect of information, anxiety and resilience on stockpiling by Muslim consumers during the COVID-19 pandemic. Journal of Islamic Marketing, 12(3), 479-497.

Lehberger, M., Kleih, A. K., & Sparke, K. (2021). Panic buying in times of coronavirus (COVID-19): Extending the theory of planned behavior to understand the stockpiling of nonperishable food in Germany. Appetite, 161, 105118.

Gong, J., Said, F., Ting, H. et al. Do Privacy Stress and Brand Trust still Matter? Implications on Continuous Online Purchasing Intention in China. Curr Psychol 42, 15515–15527 (2023). https://doi.org/10.1007/s12144-022-02857-x

From the portrayal of panic buying on newspaper, we shall also talk about how to be responsible or make it a responsible reporting, marketing , and benefit the stakeholder. I hope authors could talk a bit about responsibility for both journalists and marketers. One paper for your to refer:

Ting, H., Gong, J., Cheah, J.H.(J). and Chan, K. (2024), "Editorial: The infodemic, young consumers and responsible stakeholdership", Young Consumers, Vol. 25 No. 4, pp. 421-424. https://doi.org/10.1108/YC-04-2024-2059

Lastly, please double-check you language and i found some minor language issues and hope you can check carefully before submitting back to the journal.

Reviewers' comments:

Reviewer's Responses to Questions

**Comments to the Author**

1. Is the manuscript technically sound, and do the data support the conclusions?

Reviewer #1: Yes

Reviewer #2: Partly

2. Has the statistical analysis been performed appropriately and rigorously? 

Reviewer #1: Yes

Reviewer #2: Yes

3. Have the authors made all data underlying the findings in their manuscript fully available?

Reviewer #1: Yes

Reviewer #2: Yes

4. Is the manuscript presented in an intelligible fashion and written in standard English?

Reviewer #1: Yes

Reviewer #2: Yes

5. Review Comments to the Author

Reviewer #1: Title: The portrayal of panic-buying and stockpiling in English newspapers during Covid, a mixed method content analysis

I think this study is an interesting topic with the potential for enriching research on the portrayal of panic-buying and stockpiling in English newspapers during the Covid pandemic. Using mixed-method content analysis, the study provides valuable insights into how media coverage shaped public perception and behaviours in times of crisis. The analysis may reveal patterns in language use and narrative framing, offering an understanding of how media discourse influences societal reactions. I encourage the authors to ensure that the study maintains a clear academic focus, avoiding any potential misinterpretations or politicisation of the subject. Additionally, there are several areas where the study could be further improved to enhance its contribution.

1. Abstract

Clarify key terms and scope: I think the abstract should briefly define key concepts such as "sensationalised language" and explain the criteria used for coding articles as "negative" or "positive." This will help readers understand the methodology more clearly.

Strengthen the connection between findings and implications: After mentioning the six key themes and conclusions, the abstract should explicitly connect these insights to the potential impact on policymaking and crisis communication, offering a more direct conclusion about the relevance of the research for future crises.

Consolidate the abstract into a single cohesive paragraph: Merging the abstract into one well-structured paragraph will enhance its flow and readability, ensuring that the study's background, objectives, methodology, key findings, and implications are presented in a more concise and integrated manner, avoiding unnecessary fragmentation.

2. Keywords

Enhance Searchability: Please list your keywords to enhance the searchability of this study.

3. Introduction

Clarify the research gap earlier: I hope you can introduce the lack of media content analysis on Covid panic-buying in the UK press earlier in the introduction. This will help readers understand the study's unique contribution from the start and emphasise the research's relevance in filling this gap.

Streamline the government response section: The introduction contains many details on the government's actions and criticisms. I suggest the you consider summarising this more succinctly to avoid overwhelming the reader while still providing context for the role of media in public perception and government accountability.

Enhance the flow between media and panic-buying discussions: To strengthen the argument, you need to enchance the connection between media portrayal and panic-buying behaviour could be more clearly emphasized. Transition more smoothly between discussing the media’s role and its impact on public behaviour to keep the focus clear and coherent.

Research Questions: I strongly recommend including research questions at the end of the Introduction to better guide readers and clarify the focus of your study.

4. Literature Review

Why don’t you write a literature review for this work? I think separating the introduction from the literature review part is necessary.

5. Method

Clarify the rationale for newspaper selection: While the selection of six popular newspapers is explained, it would be helpful to clarify why these specific titles were chosen about their readership and political stance and how these factors might influence the portrayal of panic-buying, providing a more substantial justification for the sample.

Simplify technical terms for broader readability: Terms like "manifest" and "latent content" could be briefly simplified or explained in more accessible language to ensure a wider audience understands the methodology without needing specialised knowledge of content analysis.

Expand on Nexis's limitations: The paper briefly mentions Nexis's limitation of not providing visual content. Why not expand on how this limitation might have impacted the data collection process or how the scope of visual analysis would add transparency to the methodology?

6. Results

Clarify key data points: While the breakdown of article numbers is informative, the results would benefit from more precise explanations of why certain newspapers, such as The Daily Mirror, led the coverage. Expanding briefly on potential reasons for the prominence of specific papers would provide deeper insights into the reporting trends.

Link findings to the study's broader implications: After presenting each theme or data point, briefly connect the results to the larger context of media influence on public behaviour during the pandemic. This will help reinforce the study's contribution to agenda-setting theory and understanding panic-buying drivers.

7. Discussion, implication, and limitations

Clarify the impact of limitations on the study's conclusions: While the limitations are acknowledged, the discussion could benefit from elaborating on how these limitations might have influenced the results. For instance, explain how excluding social media and other forms of media might affect the generalizability of the findings regarding media influence on panic-buying behaviours.

Strengthen the linkage between findings and policy recommendations: The section mentions policy implications but could more explicitly connect the study's findings to concrete policy actions. Providing specific examples of how policymakers can utilise the insights from the study to improve communication strategies and food system resilience would enhance the practical relevance of the research.

Provide a more balanced interpretation of media influence: While the discussion focuses on the media's role in shaping panic-buying behaviour, it could also acknowledge other contributing factors, such as psychological responses to crises or government communication strategies. Incorporating a broader perspective would offer a more nuanced understanding of panic-buying phenomena.

8. Conclusion

Expand the length and detail of the conclusion: The conclusion is currently too brief and could be expanded to provide a more thorough summary of the essential findings and their implications. It should offer more specific policy recommendations and further emphasise the study's contribution to agenda-setting and framing theory while also acknowledging limitations and areas for future research. This would give the conclusion more weight and reinforce the significance of the study.

I hope that these suggestions can improve your work and make it more acceptable for the publication in this journal.

Reviewer #2: Thank you for sharing the manuscript titled “The portrayal of panic-buying and stockpiling in English newspapers during Covid, a mixed method content analysis”. Your study provides insights into media influence on public behavior during the COVID-19 pandemic. Your comprehensive analysis offers a thoughtful perspective on the interplay between media coverage and crisis management. It shows the potential areas for policy improvement. To further improve the quality of your paper, we would like to offer the following suggestions:

1.The introduction is rich in information. Perhaps you could consider dividing it into subsections (e.g., causes of panic-buying, government responses, role of the media) to enhance readability and structure. While the research background is well-described, you might want to articulate the specific aims, methods, and supporting theories more clearly. It could be beneficial to explicitly state the research purpose, questions, and significance at the end of the Introduction.

2.Regarding the theoretical framework, you might consider elaborating on the theories underpinning your study. For instance, you could clarify how the Agenda-setting theory or framing theory guides your research, and justify their application.

3.In the Methodology section, you might find it helpful to provide more detailed descriptions of content analysis methods and visual content analysis techniques. Perhaps you could strengthen the explanation of how visuals are systematically coded and how these visual elements link to your research concerns. Including specific examples of the coding and analysis process could enhance clarity and replicability.

4.You might consider adding an analytical framework to the Methodology section, offering an overview of your tiered methodology. This framework could clearly present the research focus areas addressed in the Results section and demonstrate how they connect to your methods, analysis, and discussion.

5.Regarding conclusions, I would like to read a meaningful concluding section, as well as future research perspectives.

6.Regarding language and grammar, a thorough review of the text might be beneficial.

7.Lastly, you might want to include additional bibliography where necessary and address any typographical, formatting, or referencing errors in the text.

6. PLOS authors have the option to publish the peer review history of their article (what does this mean?). If published, this will include your full peer review and any attached files.

Reviewer #1: **Yes: **Jie Zeng

Reviewer #2: No

---

## [Author Response · Author response to Decision Letter 0]

8 Nov 2024

Dear Editor, 

We have uploaded a response to reviewers alongside the manuscript and the manuscript with track changes. We have included a table in the letter which addresses each comment in turn, and is not replicable in this comment box. 

We very much look forward to your feedback, and would be delighted to have this manuscript published in PLOS1. 

Kind regards

Dayna Brackley

---

## [Decision Letter · Decision Letter 1]

21 Nov 2024

The portrayal of panic-buying and stockpiling in English newspapers during Covid, a mixed-method content analysis

PONE-D-24-41107R1

Dear Dr.Dayna Brackley,

We’re pleased to inform you that your manuscript has been judged scientifically suitable for publication and will be formally accepted for publication once it meets all outstanding technical requirements.

Kind regards,

Jiankun Gong

Academic Editor

PLOS ONE

Reviewers' comments:

Reviewer's Responses to Questions

**Comments to the Author**

1. If the authors have adequately addressed your comments raised in a previous round of review and you feel that this manuscript is now acceptable for publication, you may indicate that here to bypass the “Comments to the Author” section, enter your conflict of interest statement in the “Confidential to Editor” section, and submit your "Accept" recommendation.

Reviewer #1: All comments have been addressed

Reviewer #2: All comments have been addressed

2. Is the manuscript technically sound, and do the data support the conclusions?

Reviewer #1: Yes

Reviewer #2: Yes

3. Has the statistical analysis been performed appropriately and rigorously? 

Reviewer #1: Yes

Reviewer #2: Yes

4. Have the authors made all data underlying the findings in their manuscript fully available?

Reviewer #1: Yes

Reviewer #2: Yes

5. Is the manuscript presented in an intelligible fashion and written in standard English?

Reviewer #1: Yes

Reviewer #2: Yes

6. Review Comments to the Author

Reviewer #1: I believe this revised version has incorporated my previous feedback. However, I would like to request that you technically address the following three points:

1. Please combine your abstract paragraphs into a single cohesive paragraph, and also consider integrating some of the shorter paragraphs within the article to improve flow and readability.

2. Enhance the aesthetics of your tables to make them more legible and user-friendly.

3. The figures in your document appear to be blurry. I recommend using Plos Pace, a website specializing in figure optimization, to convert your blurry figures into high-resolution 300 dpi TIFF files.

Reviewer #2: Dear Authors，

Congratulations! I am pleased to inform you that your paper can accepted for publication. Before final submission, I would kindly encourage you to carefully review of the manuscript's formatting consistency, reference accuracy, and overall language clarity. This final check will ensure your excellent work meets the highest publication standards.

Thank you for your work.

Best regards,

7. PLOS authors have the option to publish the peer review history of their article (what does this mean?). If published, this will include your full peer review and any attached files.

Reviewer #1: No

Reviewer #2: No

---

## [Editor Report · Acceptance letter]

25 Nov 2024

PONE-D-24-41107R1 

PLOS ONE

Dear Dr. Brackley, 

I'm pleased to inform you that your manuscript has been deemed suitable for publication in PLOS ONE. Congratulations! Your manuscript is now being handed over to our production team.

Kind regards, 

on behalf of

Dr. Jiankun Gong 

Academic Editor

PLOS ONE